# Gas phase synthesis of [4]-helicene

Long Zhao[1], Ralf I. Kaiser [1], Bo Xu[2], Utuq Ablikim[2], Wenchao Lu[2], Musahid Ahmed [2], Mikhail M. Evseev[3], Eugene K. Bashkirov[3], Valeriy N. Azyazov[3], Marsel V. Zagidullin[3], Alexander N. Morozov[4], A. Hasan Howlader[4], Stanislaw F. Wnuk [4], Alexander M. Mebel[3,4], Dharati Joshi[5], Gregory Veber[5] & Felix R. Fischer[5,6,7]

A synthetic route to racemic helicenes via a vinylacetylene mediated gas phase chemistry involving elementary reactions with aryl radicals is presented. In contrast to traditional synthetic routes involving solution chemistry and ionic reaction intermediates, the gas phase synthesis involves a targeted ring annulation involving free radical intermediates. Exploiting the simplest helicene as a benchmark, we show that the gas phase reaction of the 4-phenanthrenyl radical ($[C_{14}H_9]^{\bullet}$) with vinylacetylene ($C_4H_4$) yields [4]-helicene ($C_{18}H_{12}$) along with atomic hydrogen via a low-barrier mechanism through a resonance-stabilized free radical intermediate ($C_{18}H_{13}$). This pathway may represent a versatile mechanism to build up even more complex polycyclic aromatic hydrocarbons such as [5]- and [6]-helicene via stepwise ring annulation through bimolecular gas phase reactions in circumstellar envelopes of carbon-rich stars, whereas secondary reactions involving hydrogen atom assisted isomerization of thermodynamically less stable isomers of [4]-helicene might be important in combustion flames as well.

[1] Department of Chemistry, University of Hawaii at Manoa, Honolulu, HI 96822, USA. [2] Chemical Sciences Division, Lawrence Berkeley National Laboratory, Berkeley, CA 94720, USA. [3] Samara National Research University, Samara 443086, Russia. [4] Department of Chemistry and Biochemistry, Florida International University, Miami, FL 33199, USA. [5] Department of Chemistry, University of California, Berkeley, CA 94720, USA. [6] Materials Sciences Division, Lawrence Berkeley National Laboratory, Berkeley, CA 94720, USA. [7] Kavli Energy Nano Sciences Institute at the University of California Berkeley and the Lawrence Berkeley National Laboratory, Berkeley, CA 94720, USA. Correspondence and requests for materials should be addressed to R.I.K. (email: ralfk@hawaii.edu)

During the last decades, helicenes—ortho-fused polycyclic aromatic hydrocarbons (PAHs), in which benzene building blocks are annulated at an angle of 60° to form helically-shaped molecules[1]—have received considerable attention from the organic[2], and physical chemistry[3], and material science communities[4,5] due to their unique features in optics (chiroptical activity[6], nonlinear optics[3], and circular polarization[7]) and chiral sensing (chemical sensors)[8,9] along with exceptional properties in organocatalysis[4,10,11] and distinctive molecular structures (Fig. 1)[12]. Considering the molecular structure of helicenes, the backbone twists in opposite directions due to the steric hindrance between the terminal rings. Within the helicene series, the dihedral angles between both terminal benzene moieties rises from [4]-helicene (26°) via [5]-helicene (46°) to [6]-helicene (58°) before it drops in [7]-helicene (30°). Helicenes are distinguished for their chirality despite missing asymmetric carbon atoms with the chirality developing from the handedness of the helix. Clockwise and counterclockwise helices are non-superimposable as the result of their axial chirality with a left- and right-handed helices being defined by minus (**M**) and plus (**P**)[13]. The racemization mechanism depends on the size of the [n]-helicene involving a concerted process for n = 4–7, but a multistep mechanism for larger units. A single transition state with $C_{2v}$ (n = 4) and $C_s$ (n = 5–7) symmetries and barriers

increasing from about 17 kJ mol$^{-1}$ via 102 kJ mol$^{-1}$ and 154 kJ mol$^{-1}$ to 176 kJ mol$^{-1}$ connects the **M** and **P** forms[2,3,14–16].

The first helicene synthesis reported by Meisenheimer more than a century ago[17] describes the formation of two aza-helicenes in the reduction of 2-nitronaphthalene with metallic zinc under basic conditions. The first carbohelicene, [5]-helicene, was synthesized only 15 years later by Weitzenböck and Klingler[18] through copper-induced radical cyclization of 2-diazostilbenes. The larger homolog [6]-helicene[19] was first obtained through intramolecular Friedel-Crafts cyclization of an acid chloride followed by reductive deoxygenation and rearomatization. While early synthesis of helicenes took advantage of an intramolecular oxidative photocyclization of readily accessible stilbenes[20], a great variety of modern transformations have been applied. Among the most versatile are [2 + 4][21–27] and [2 + 2 + 2] cycloaddition reactions[28–32], a variety of transition-metal catalyzed cross-coupling reactions[33,34], and more recently ring closing metathesis[35,36]. Examples for radical transformations, such as the seminal copper catalyzed cyclization reported by Meisenheimer are rare, often limited by functional group tolerance and the formation of linear side products resulting from poor *E/Z* regioselectivity in the cyclization step[37–39].

Here, we reveal a versatile route to form helicenes via a directed, vinylacetylene mediated gas phase chemistry. In contrast

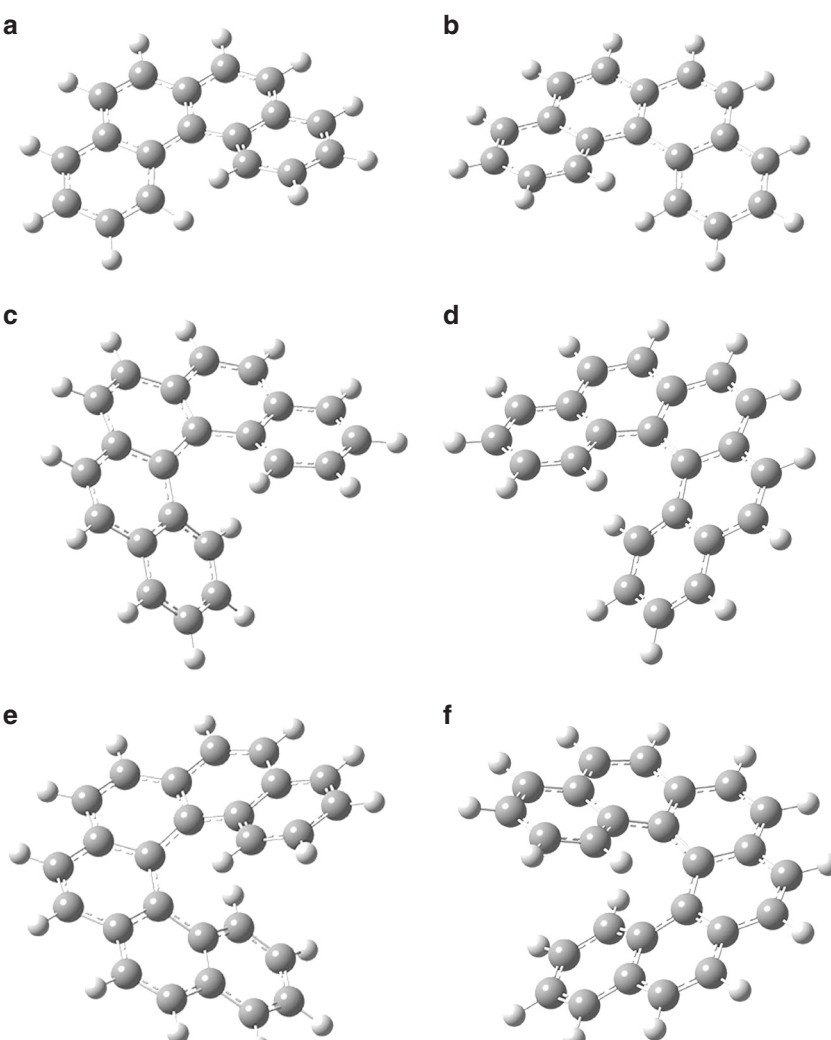

**Fig. 1** Molecular structures of helicenes. **a** (*M*)-[4]-helicene; **b** (*P*)-[4]-helicene; **c** (*M*)-[5]-helicene; **d** (*P*)-[5]-helicene; **e** (*M*)-[6]-helicene and **f** (*P*)-[6]-helicene

to the aforementioned routes following solution chemistry and often ionic reaction intermediates[28,35,40,41], the innovative gas phase synthesis encompasses low-barrier reactions through targeted, stepwise ring expansion mechanisms involving free radical reaction intermediates. Exploiting the simplest helicene as a benchmark, we reveal the hitherto unknown gas phase chemistry synthesizing [4]-helicene ($C_{18}H_{12}$; 228 amu) along with atomic hydrogen (1 amu) via the bimolecular reaction of the 4-phenanthrenyl radical ($[C_{14}H_9]^{\bullet}$; 177 amu) with vinylacetylene ($C_4H_4$; 52 amu) (reaction (1)). Our combined experimental and *ab initio* study exposes the prototype of a low-barrier reaction mechanism leading to the facile formation of the simplest helicene. [4]-helicene—an 18-$\pi$-electron aromatic molecule—was formed via a directed synthesis through molecular mass growth involving the reaction of an aromatic radical—4-phenanthrenyl—with a single vinylacetylene molecule embracing a low-barrier ring-expansion via a resonance-stabilized free radical (RSFR) intermediate ($C_{18}H_{13}$). This pathway represents a versatile reaction mechanism to build up even more complex helicenes such as [5]- and [6]-helicene via stepwise ring expansion through bimolecular gas phase reactions of an aryl radical with vinylacetylene. In more complex environments such as combustion flames, this pathway competes with a secondary reactions involving a hydrogen atom assisted isomerization of the thermodynamically less stable (E)-4-(but-1-en-3-yn-1-yl) phenanthrene isomer. Briefly, a chemical reactor was exploited to synthesize [4]-helicene via the elementary gas phase reaction of the 4-phenanthrenyl radical ($[C_{14}H_9]^{\bullet}$) with vinylacetylene ($C_4H_4$). The reaction products were probed isomer-specifically in a molecular beam through fragment-free photoionization of the neutral products via tunable vacuum ultraviolet (VUV) light followed by detection of the ionized molecules in a reflectron time-of-flight mass spectrometer (Re-TOF-MS) (Methods).

$$[C_{14}H_9]^{\cdot} + C_4H_4 \rightarrow C_{18}H_{12} + H^{\cdot} \qquad (1)$$

## Results

**Mass spectra results.** Figure 2a displays a representative mass spectrum recorded at a photoionization energy of 9.50 eV for the reaction of the 4-phenanthrenyl radical with vinylacetylene; reference spectra were also collected by substituting the vinylacetylene reactant with non-reactive helium carrier gas (Fig. 2b). A comparison of these data sets provides persuasive evidence on the synthesis of a molecule with the molecular formula $C_{18}H_{12}$ (228 amu) in the 4-phenanthrenyl–vinylacetylene reaction (Fig. 2a), which is clearly absent in the control study (Fig. 2b). Accounting for the molecular weight of the reactants and the products, it is evident that the $C_{18}H_{12}$ isomer(s) along with atomic hydrogen is the result of the bimolecular reaction of the 4-phenanthrenyl radical with vinylacetylene (reaction (1)). A signal for $C_{16}H_{10}$ (202 amu) is likely attributed to the reaction of 4-phenanthrenyl ($[C_{14}H_9]^{\bullet}$; 177 amu) with acetylene ($C_2H_2$; 26 amu) and can be identified as pyrene and ethynylphenanthrene[42] (Supplementary Fig. 1). Finally, signals at mass-to-charge ratios ($m/z$) of 259 ($C_{13}^{13}CH_9^{81}Br^+$), 258 ($C_{14}H_9^{81}Br^+$), 257($C_{13}^{13}CH_9^{79}Br^+$), 256 ($C_{14}H_9^{79}Br^+$), 179 ($C_{13}^{13}CH_{10}^+$), 178 ($C_{14}H_{10}^+$), 177 ($C_{14}H_9^+$/ $C_{14}^{13}CH_8^+$), and 176 ($C_{14}H_8^+$) can be detected in both the 4-phenanthrenyl–vinylacetylene and the 4-phenanthrenyl–helium experiments. Consequently, these molecules cannot be attributed to the reaction of 4-phenanthrenyl with vinylacetylene. Signal at $m/z = 259$–256 can be associated with the non-pyrolyzed 4-bromophenanthrene precursor, whereas ion counts at $m/z = 178$ and 179 are reflective of phenanthrene and $^{13}$C-phenanthrene generated through hydrogen abstraction by the 4-phenanthrenyl radical or hydrogen atom addition to this radical. Finally, a signal at $m/z = 176$ and 177 is related to distinct phenanthryne isomers ($m/z = 176$) along with the unreacted 4-phenanthrenyl radical ($[C_{14}H_9]^{\bullet+}$; $m/z = 177$) (Supplementary Fig. 1).

**Photoionization efficiency spectra.** Having identified hydrocarbon molecule(s) of the molecular formula $C_{18}H_{12}$ synthesized in the elementary reaction of the 4-phenanthrenyl radical with vinylacetylene, we elucidate the nature of the structural isomer(s)

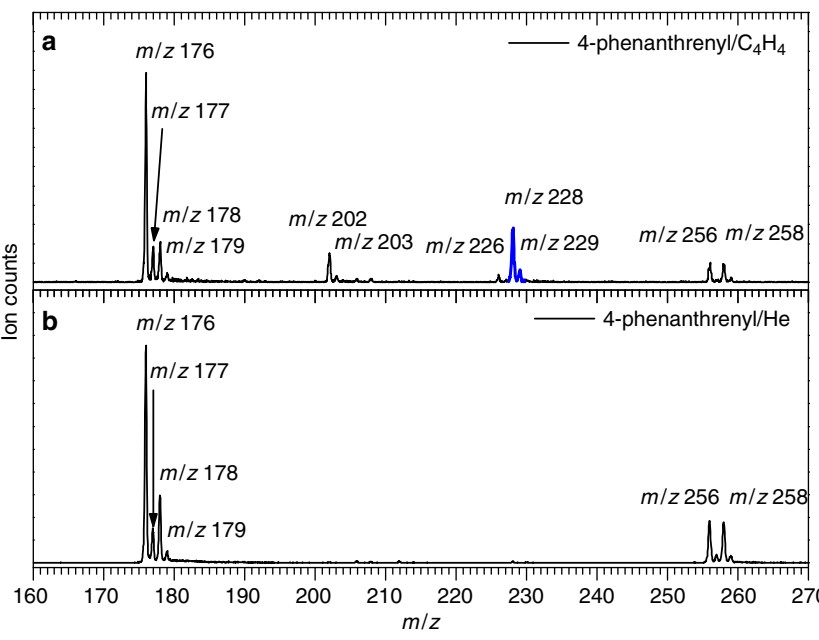

**Fig. 2** Comparison of photoionization mass spectra recorded at a photoionization energy of 9.50 eV. **a** 4-phenanthrenyl ($[C_{14}H_9]^{\bullet}$) - vinylacetylene ($C_4H_4$) system; **b** 4-phenanthrenyl ($[C_{14}H_9]^{\bullet}$) - helium (He) system. The mass peaks of the newly formed $C_{18}H_{12}$ ($m/z = 228$) species along with the $^{13}$C-substituted counterparts ($m/z = 229$) are highlighted in blue

   

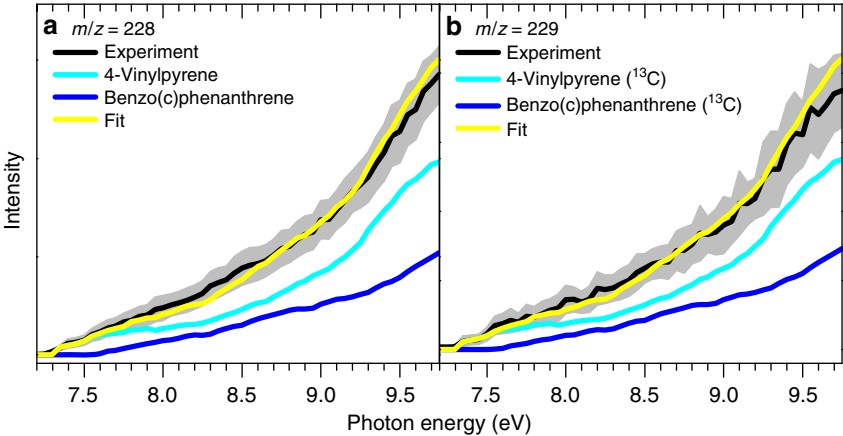

**Fig. 3** Photoionization efficiency (PIE) curves relevant to the formation of [4]-helicene. **a** $m/z = 228$ and **b** $m/z = 229$. Black: experimentally derived PIE curves; blue: [4]-helicene (benzo[c]phenanthrene) reference PIE curve; cyan: 4-vinylpyrene PIE curve; yellow: overall fit. The overall error bars (gray area) consist of two parts: ±10% based on the accuracy of the photodiode and a 1 $\sigma$ error of the PIE curve averaged over the individual scans

formed in this reaction. This necessitates a thorough examination of the corresponding photoionization efficiency (PIE) curve, which reports the intensity of the ion at $m/z$ of 228 ($C_{18}H_{12}^+$) as a function of the photon energy from 7.20 eV–9.75 eV (Fig. 3a). These data have to be fit with a linear combination of distinct reference PIE curves of discrete $C_{18}H_{12}$ isomers: [4]-helicene (benzo[c]phenanthrene), 4-vinylpyrene, chrysene, benz[a] anthracene, triphenylene, 4-((E)-but-1-en-3-yn-1-yl)phenanthrene, and 4-(but-3-en-1-yn-1-yl)phenanthrene (Supplementary Fig. 2). More explicitly, a linear combination of two reference curves is required to replicate the experimentally derived PIE curve for $m/z$ of 228 (black): [4]-helicene (blue) and 4-vinylpyrene (red). The experimental and reference PIE curves for 4-vinylpyrene depict an onset of the ion signal at 7.30 ± 0.05 eV. However, at photoionization energies higher than 7.60 eV, the PIE reference curve of 4-vinylpyrene cannot fit the experimental PIE curve alone, and ion counts are clearly lacking. To account for this mismatch, an incorporation of the PIE curve of [4]-helicene is required. The overall fit (black) of the ion counts consists of [4]-helicene (33 ± 10%) and of 4-vinylpyrene (67 ± 10%). It should be noted that inclusions of any contributions from 4-((E)-but-1-en-3-yn-1-yl)phenanthrene and 4-(but-3-en-1-yn-1-yl) phenanthrene do not improve the fit of the experimental data. This indicates that these isomers can be produced only at levels within the experimental error limits of the ion counts. We emphasize here that the aforementioned contributions to the fit of the experimental PIE curve do not represent the product branching ratios since absolute ionization cross sections are not available for any of the $C_{18}H_{12}$ isomers. A more detailed discussion on the computed product branching ratios based on computational fluid dynamics (CFD) simulations coupled with kinetic modeling of chemical reactions inside the reactor is given in the Supporting Information (Supplementary Note 2, Supplementary Fig. 5, Supplementary Tables 1–4). It is important to highlight that the PIE curve recorded at $m/z = 229$ (Fig. 3b) matches after scaling the PIE curve of $m/z = 228$. Consequently, the data at $m/z = 229$ can be linked with [13]C substituted isomer ($C_{17}^{13}CH_{12}$) of [4]-helicene and 4-vinylpyrene with an overall abundance of close to 20% to account for the 1.1% [13]C natural abundance. It is vital to highlight that the PIE curves of the $C_{18}H_{12}$ isomers are specifically linked to each specific isomer underlining that the co-existence of additional isomers in the molecular beam except [4]-helicene and 4-vinylpyrene would significantly alter the shape of the PIE; therefore, alternative $C_{18}H_{12}$ isomers can be eliminated to contribute to $m/z = 228$.

Consequently, we conclude that only [4]-helicene and 4-vinylpyrene account for the observed signal at $m/z = 228$ within our error limits. The latter are based on the 1 $\sigma$ error of the PIE curve averaged over multiple scans and the ±10% error stated by the manufacturer as the accuracy of the photodiode.

## Discussion

This study provides evidence that the simplest representative of the class of helicenes, i.e., [4]-helicene, can be formed in a directed synthesis through the gas phase elementary reaction of the 4-phenanthrenyl radical with vinylacetylene. Having identified [4]-helicene along with its 4-vinylpyrene isomer, we are attempting now to elucidate the underlying reaction mechanisms. Therefore, the experimental data are merged with electronic structure calculations on the $C_{18}H_{12}$ and $C_{18}H_{13}$ potential energy surfaces (PESs) (Fig. 4, Supplementary Fig. 4) (Methods). The calculations reveal that the vinylacetylene molecule approaches the 4-phenenthrenyl radical leading to the formation of two weakly stabilized distinct van-der-Waals complexes [1] and [2]; in each complex, the radical center at the C4 carbon atom is directed to the $H_2C=$ or HCC moiety of the vinylacetylene reactant at the C1 and C4 carbon atom, respectively. These complexes can isomerize through addition of the radical center to the carbon–carbon double and triple bond, respectively, forming a covalent carbon–carbon bond and intermediates [3] and [4] after passing through transition states located 14 and 17 kJ mol⁻¹ above the van-der-Waals complex, respectively. In our earlier work[43,44], the addition of the smallest aryl radical, phenyl, to the $H_2C=$ moiety at the C1 carbon atom of vinylacetylene was found to be peculiar. There, a barrier to addition exists, but the corresponding transition state lies lower in energy than the separated reactants. Therefore, a barrier is present, however, the inherent transition state is located below the energy of the separated reactants and hence defined as a submerged barrier. The situation appears to be different for the 4-phenanthrenyl radical, where the barrier to form the carbon–carbon covalent bond is not submerged, but instead resides 6 to 9 kJ mol⁻¹ above the separated reactants according to the G3(MP2,CC) and higher-level CCSD (T)/CBS calculations (Methods), respectively. The increase of the entrance barrier height in the 4-phenanthrenyl plus vinylacetylene reaction as compared to phenyl plus vinylacetylene is caused by the steric repulsion with the bay hydrogen atom in 4-phenanthrenyl, which hinders the approach of the $H_2C=$ group toward the radical site. The isomerization of [2] to [4] via

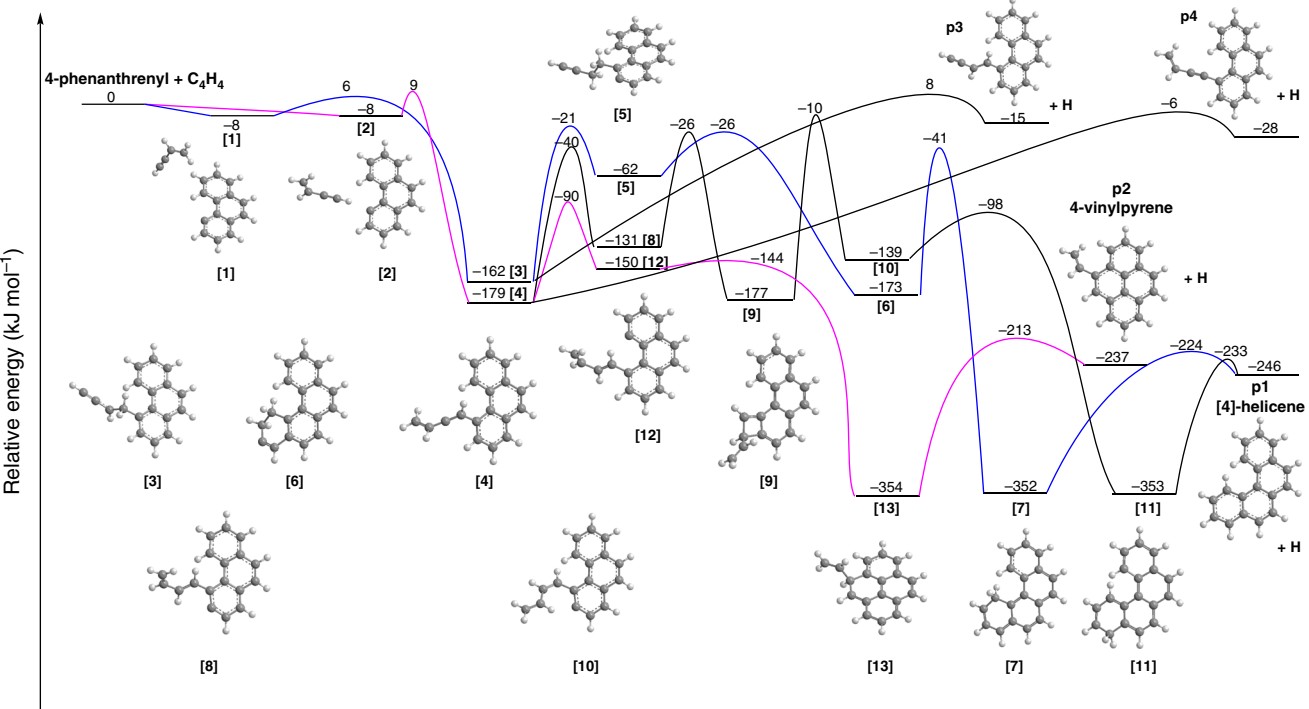

**Fig. 4** Potential energy surface (PES) for the 4-phenanthrenyl [C₁₄H₉]• reaction with vinylacetylene (C₄H₄). This PES was calculated at the G3(MP2,CC)// B3LYP/6-311G(d,p) level of theory for the channels leading to [4]-helicene **p1**, 4-vinylpyrene **p2**, 4-((E)-but-1-en-3-yn-1-yl)phenanthrene **p3**, and 4-(but-3-en-1-yn-1-yl)phenanthrene **p4**. The relative energies are given in kJ mol⁻¹

addition of the 4-phenanthrenyl radical to the HCC moiety proceeds via the transition state residing 9 kJ mol⁻¹ above the energy of the separated reactants at the G3(MP2,CC)//B3LYP/6-311G(d,p) level of theory, which is also higher than the corresponding energy in the phenyl plus vinylacetylene system, 5 kJ mol⁻¹, but the steric hindrance for the HCC group approach is less than for the bulkier H₂C= moiety.

The C₁₈H₁₃ intermediate [3] rearranges via a hydrogen shift from the C3 carbon atom of the phenanthrenyl moiety to the former vinylacetylene reactant forming intermediate [5] by switching the unpaired electron effectively from the side chain to the aromatic ring, which subsequently follows ring closure through a barrier of only 36 kJ mol⁻¹ to intermediate [6]. The latter already mirrors the carbon skeleton of the target [4]-helicene and undergoes a hydrogen shift from the methylene group to the carbene center providing intermediate [7]. A closer look at this structure reveals that only an atomic hydrogen loss accompanied by aromatization is required to form [4]-helicene (**p1**) in an overall exoergic reaction (246 kJ mol⁻¹). This hydrogen elimination proceeds almost perpendicularly to the molecular plane and passes a tight exit transition state located 22 kJ mol⁻¹ above the separated products. Note that the existence of this tight exit transition state is reasonable because in the reverse reaction, the hydrogen atom has to add to a closed shell aromatic molecule, and the bond formation requires a significant change in electron density. This barrier to addition is of comparable magnitude as of 37 kJ mol⁻¹ and 26 kJ mol⁻¹ determined for the addition of atomic hydrogen to benzene and naphthalene—two benchmark aromatic systems[45,46].

What is the fate of the second C₁₈H₁₃ intermediate [4]? The computations reveal that this intermediate can undergo two competing rearrangement pathways leading either to [4]-helicene or 4-vinylpyrene. The channel producing [4]-helicene is multistep and rather demanding both in terms of energy and entropy. It

begins from a [1,4]-H shift from the attacked ring in the phenanthrene core to the bare carbon atom of the side chain in [4] forming [8] via a barrier of 139 kJ mol⁻¹. Intermediate [8] cannot undergo immediate ring-closure because its side chain features a *trans* conformation with respect to the C=C double bond. Rotation around the double bond is not feasible and hence the *trans-cis* isomerization to [10] occurs in two steps via intermediate [9] and involves a closure and opening of a four-member ring. However, the barriers along the [8] → [9] → [10] path are rather high. Next, [10] undergoes facile ring closure to [11] and a hydrogen atom loss in the latter produces [4]-helicene via an exit barrier of only 13 kJ mol⁻¹. The alternative reaction channel starting from [4] is much more favorable. It begins with a [1,6]-H shift through the bay region of the phenanthrene core to the side chain forming [12] over a relatively low-barrier of 89 kJ mol⁻¹. [12] is easily subjected to a six-member ring closure in the bay to form [13] and a hydrogen atom loss from [13] finally produces 4-vinylpyrene (**p2**)—a structural isomer of [4]-helicene—in an overall exoergic reaction (237 kJ mol⁻¹) with an exit barrier of 24 kJ mol⁻¹. Consequently, the computational prediction of the formation of two structural isomers of C₁₈H₁₂— [4]-helicene and 4-vinylpyrene—is well matched by our experimental studies and directly reflects two distinct entrance channels to reaction via van-der-Waals complexes [1] and [2] and hence regioselective reaction dynamics leading to two discrete structural isomers thus defining a benchmark of a molecular mass growth process to PAHs. It should be noted that the initial covalently bound intermediates [3] and [4] can undergo hydrogen atom losses to form 1- and 4-phenanthrenyl-vinylacetylene isomers **p3** and **p4**, respectively. These pathways are noticeable less favorable energetically than the formation pathways of [4]-helicene and 4-vinylpyrene—more so that in the prototype phenyl plus vinylacetylene reaction—but may play a role at high temperatures due to a favorable entropic factor. To explore this possibility, we also

**Fig. 5** Mechanism. Schematic presentation of a vinylacetylene mediated helicene propagation to [5]-helicene and [6]-helicene starting from [4]-helicene (benzo[c]phenanthrene)

conducted statistical Rice-Ramsperger-Kassel-Marcus Master Equation (RRKM-ME) calculations of product branching ratios at 1400 K (Supplementary Note 1). These calculations infer that in complex systems such as in combustion flames, [4]-helicene can be also formed via secondary reactions involving a hydrogen atom assisted isomerization of the (E)-4-(but-1-en-3-yn-1-yl) phenanthrene isomer (**p3**, Supplementary Fig. 4); however, in environments where bimolecular reactions dominate such as in circumstellar envelopes of carbon stars, secondary reactions are absent, and [4]-helicene is formed via the elementary reaction of the 4-phenanthrenyl radical ($[C_{14}H_9]^•$) with vinylacetylene ($C_4H_4$).

The facile route to synthesize [4]-helicene ($C_{18}H_{12}$) through the reaction of the 4-phenanthrenyl radical ($[C_{14}H_9]^•$) with vinylacetylene ($C_4H_4$) represents a versatile pathway that could in principle be extended to higher helicenes. Our mechanistic studies reveal that the key step of the reaction is a low-barrier benzannulation through a resonance-stabilized free radical (RSFR) intermediate that leads to the simplest helicene. This pathway involving a bimolecular collision can serve as an alternative to other, commonly accepted gas phase reactions such as the traditional hydrogen abstraction–acetylene addition (HACA) route[47,48]. HACA, which has been invoked in an attempt to unravel the formation of PAHs in high-temperature environments such as in combustion flames[49–51] and in outflows of the carbon-rich asymptotic giant branch (AGB) stars such IRC + 10216[52], cannot lead to helicenes under single collision conditions. This pathway would rather lead to bay closure, but not to benzannulation as verified in the biphenyl ($C_{12}H_9$)[51]—acetylene and 4-phenanthrenyl ($[C_{14}H_9]^•$)—acetylene systems (Supplementary Fig. 3)[42]. Consequently, PAHs as complex as [4]-helicene ($C_{18}H_{12}$) can be formed via a bimolecular reaction in circumstellar envelopes of dying carbon stars. However, in combustion flames such as of benzene[53,54], where free hydrogen atoms are ubiquitous, secondary reactions via a hydrogen atom assisted isomerization of the (E)-4-(but-1-en-3-yn-1-yl)phenanthrene isomer likely also play an important role in the formation of [4]-helicene. In circumstellar environments, starting from [4]-helicene, propagation via hydrogen loss by photodissociation followed by reaction with vinylacetylene provides an exceptional route to [5]-helicene and eventually to [6]-helicene (Fig. 5) thus supplying a directed, stepwise synthesis of racemic, helically-shaped three-dimensional nanostructures via elementary neutral-neutral reactions.

## Methods

**Experimental**. The experiments were carried out at the Advanced Light Source (ALS) at the Chemical Dynamics Beamline (9.0.2.) utilizing a high-temperature chemical reactor consisting of a resistively-heated silicon carbide (SiC) tube of 20 mm length and 1 mm inner diameter[50,51,55]. This reactor is integrated into a molecular beam apparatus operated with a Wiley-McLaren reflectron time-of-flight mass spectrometer (Re-TOF-MS). This setup is designed to explore elementary chemical reactions to mimic PAH growth in situ through the reaction of aromatic radicals. In detail, 4-phenanthrenyl radicals $[C_{14}H_9]^•$ were generated in situ via pyrolysis of the 4-bromophenanthrene precursor ($C_{14}H_9Br$) seeded in vinylacetylene/helium (5% $C_4H_4$; 95% He; Airgas) carrier gas at a pressure of 300 Torr in the entrance of the reactor. Accurate estimation of the concentration of $C_{14}H_9Br$ in the incipient molecular beam is difficult because the vapor pressure of this precursor is not known but the results of the computational fluid dynamics simulations and kinetic modeling (Supplementary Note 2, Supplementary Fig. 5 and Supplementary Tables 1–4) are consistent with experiment at the level of $C_{14}H_9Br$ concentration of 1–5%. The temperature of the SiC tube was monitored using a Type-C thermocouple and was operated at 1400 ± 10 K. At this temperature, 4-bromophenanthrene dissociates to the 4-phenanthrenyl radical plus atomic bromine in situ followed by the reaction of the aromatic radical with vinylacetylene. The reaction products were expanded supersonically and passed through a 2-mm diameter skimmer located 10 mm downstream of the reactor and enter the main chamber, which houses the Re-TOF-MS. The products within the supersonic beam were then photoionized in the extraction region of the mass spectrometer by exploiting quasi-continuous tunable synchrotron vacuum ultraviolet (VUV) light. VUV single photon ionization is a fragment-free ionization technique and is often described as a 'soft' ionization method[56,57] compared to the harsher conditions of electron impact ionization, the latter leading to excessive fragmentation of the parent ion. The ions formed via soft photoionization in the present experiments are extracted and introduced onto a microchannel plate detector through an ion lens. Photoionization efficiency (PIE) curves, which report ion counts as a function of photon energy from 7.20 eV to 9.75 eV with a step interval of 0.05 eV at a well-defined mass-to-charge ratio (m/z), are generated by integrating the signal collected at the specific m/z for the species of interest and normalized to the photon flux. The residence time in the reactor tube under our experimental condition are few hundreds of μs[58,59]. Reference (control) experiments were also conducted by expanding neat helium carrier gas with the 4-bromophenanthrene precursor into the resistively-heated silicon carbide tube. No signals, which can be associated with the [4]-helicene of 4-vinylpyrene molecules at m/z = 228 or 229, was observed in these control experiments. Reference PIE curves of [4]-helicene, 4-vinylpyrene, (E)-4-(but-1-en-3-yn-1-yl)phenanthrene, and 4-(but-3-en-1-yn-1-yl)phenanthrene were measured in the present work. [4]-Helicene (**p1**) was purchased from Sigma-Aldrich (98%). The syntheses of 4-vinylpyrene (**p2**), (E)-4-(but-1-en-3-yn-1-yl)phenanthrene (**p3**), and 4-(but-3-en-1-yn-1-yl) phenanthrene (**p4**) are described in the Supplementary Information (Supplementary Note 3, Supplementary Figs 6–12, and Supplementary Schemes 1–3).

**Theoretical calculations**. The energies and molecular parameters of the local minima and transition states involved in the reaction were computed at the G3 (MP2,CC)//B3LYP/6-311G(d,p) level of theory[60–62] with a chemical accuracy of 3–6 kJ mol$^{-1}$ for the relative energies and 0.01–0.02 Å for bond lengths as well as 1–2° for bond angles[62]. Additional higher-level calculations for the [1]–[3] transition state were performed using at the CCSD(T) level in the complete basis set (CBS) limit involving explicitly-correlated CCSD(T)-F12b and MP2-F12

calculations[63] with Dunning's correlation-consistent basis sets[64], where the CCSD (T)/CBS total energies of the reactants and the transition state were approximated as follows:

$$E(CCSD(T)/CBS) = E(CCSD(T)-F12b/cc-pVDZ-f12)$$
$$+ E(MP2/cc-pVTZ-f12)$$
$$- E(MP2/cc-pVTZ-f12).$$

The GAUSSIAN 09[63] and MOLPRO 2010 program packages[64] were utilized for the ab initio calculations. The details of the statistical RRKM-ME calculations of the reaction rate constants and computational fluid dynamics simulations of the gas flow and chemical kinetics in the microreactor are provided in the Supplementary Information (Supplementary Notes 1 and 2, Supplementary Tables 1–4 and Supplementary Fig. 5).

## Data availability
The data that support the plots within this paper and other findings of this study are available from the corresponding author upon reasonable request.

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

## Acknowledgements

This work was supported by the US Department of Energy, Basic Energy Sciences DE-FG02-03ER15411 (experimental studies) DE-FG02-04ER15570 (computational studies and synthesis of 4-vinylpyrene, (*E*)-4-(but-1-en-3-yn-1-yl)phenanthrene, and 4-(but-3-en-1-yn-1-yl) phenanthrene) and DE-SC0010409 (synthesis of precursor molecules) to the University of Hawaii, Florida International University and the University of California Berkeley, respectively. B.X. U.A., W.L. and M.A. are supported by the Director, Office of Science, Office of Basic Energy Sciences, of the U.S. Department of Energy under Contract No. DE-AC02-05CH11231, through the Gas Phase Chemical Physcis Program, Chemical Sciences Division. The Advanced Light Source is also supported through the same contract. Ab initio calculations of the $C_{18}H_{13}$ PES relevant to the reaction of 4-phenanthrenyl radical with vinylacetylene and simulations of the gas flow and kinetics in the microreactor at Samara University were supported by the Ministry of Education and Science of the Russian Federation under Grant No. 14.Y26.31.0020. A.H. H. is supported by the Presidential Fellowship from FIU.

## Author contributions

R.I.K. designed the experiment; D.J., G.V. and F.R.F. synthesized the reactant; A.H.H. and S.F.W. synthesized the calibration compounds; L.Z., B.X., U.A. and W.L. carried out the experimental measurements; M.A. supervised the experiment; L.Z. performed the data analysis; M.E., E.K.B., V.N.A., M.V.Z., A.N.M. and A.M.M. carried out the theoretical analysis; R.I.K., A.M.M. and M.A. discussed the data; R.I.K. and A.M.M. wrote the manuscript.
