## [Peer Review File · Nature Communications]

Reviewers' comments:

Reviewer #1 (Remarks to the Author):

In their manuscript, the authors address the "Gas Phase Synthesis of [4]-Helicene via a Versatile Vinylacetylene-Mediated Free Radical Ring Annulation". They present their finding as " A universal synthetic route to racemic helicenes". This is not at all what is demonstrated in the manuscript since only the obtention of benzo[c]phenanthrene, i.e. [4]helicene, is shown and there is no evidence at all that the same mechanism applies to longer helicenes. Indeed, regioselective aspects together with steric hindrance issues may be encountered.

The authors have clearly a strong expertise in the domains of PAHs and on the techniques they describe in this manuscript both experimentally (gas phase reactions, photoionization mass spectrometry) and theoretically (potential energy calculations). However, I found the manuscript very technical, both from the experimental and from the theoretical point of view, and not general enough to deserve publication in Nature Communications. I would rather suggest a more specific journal. Of course the same work on the case study carbo[6]helicene would have had a bigger impact.

The authors mention "stereo selective reaction dynamics" when they discuss the "computational prediction of the formation of two structural isomers of C₁₈H₁₂ –" I think that they actually meant regioselective because the two isomers come from the cyclization at different positions.

The references given for the helicenes are not really appropriate. For example, the authors give references regarding catalysis or azahelicenes but more importantly there are many articles on helical PAHs, which, in my opinion, are more related to the present work.

Reviewer #3 (Remarks to the Author):

This is an interesting paper and certainly deserves to be published in Nature Communication.

What are the major claims of the paper? Are they novel and will they be of interest to others in the community and the wider field? The authors claim that they identified a gas-phase reaction route

towards 4-helicene. This is important and new for several reasons, including that until now only liquid-phase reaction mechanisms for 4-helicene synthesis exist.

Is the work convincing, and if not, what further evidence would be required to strengthen the conclusions?

Yes, given the current knowledge of gas-phase chemistry and with the theoretical evidence provided in this paper, the results are convincing. However, the discussion about the exclusion of different C₁₈H₁₂ isomers is rather weak. Again, I don't doubt their results, it's just that their arguments are pretty weak for excluding other conceivable C₁₈H₁₂ isomers.

Do you feel that the paper will influence thinking in the field? Yes, I feel that people will include this reaction in their reaction mechanisms to explore the importance of this reaction.

Comment on the ability of a researcher to reproduce the work, given the level of detail provided: Yes, all necessary details are provided.

Before final publication, I would suggest the authors taking the additional suggestions into account:

Have the authors considered other potential pathways leading to the targeted intermediate. The mass spec at the low photon energies is rather misleading because many important intermediates (such as acetylene) are not present. What else is happening in the reactor? Where does the acetylene come from and what role does it play in the reaction network?

Referee report for Nature Communications manuscript NCOMMS-18-21302

The manuscript is very well written; in particular, the introductory page is excellent. The conclusions are fascinating. It sounds a wonderful story that may be true.

But I have some concerns and few comments are therefore in order.

1) Only one typo spotted: in the Experimental Methods, 5 lines before the bottom of the section, “of 4-vinylpyrene” should be “or 4-vinylpyrene”.

2) In the last five lines before the Conclusion the isomer products p3 and p4 are mentioned, but they are not depicted in Figure 4 of the main text. They are instead depicted in Figure 4 of the Supplementary Information, but this is not specified (so one cannot see where they are while reading the main text). I suggest to include the pathways to isomers p3 and p4 also in Figure 4 of the main text.

3) My main concerns will actually start from the above mentioned five lines :

It should be noted that the initial covalently bound intermediates **[3]** and **[4]** can undergo hydrogen atom losses to form 1- and 4-phenanthrenyl-vinylacetylene isomers **p3** and **p4**, respectively. These pathways are noticeable less favorable energetically than the formation pathways of [4]-helicene and 4-vinylpyrene – more so that in the prototype phenyl plus vinylacetylene reaction – but may play a role at high temperatures due to a favorable entropic factor.

Having said that I am very impressed by this manuscript, I have to say that I am also puzzled by several aspects of this study, and in particular by the approach used to reach the conclusions. We all know that the unambiguous proof that the bimolecular reaction $C_{14}H_9 + C_4H_4$ leads to $H + C_{18}H_{12}$ (4-helicene) can only come from experiments under single-collision conditions, as feasible using the crossed molecular beam (CMB) scattering technique. Prof. Kaiser has stressed this since the beginning of his career. Prof. Kaiser is one of the leading players in CMB experiments and has built his reputation by performing very nice and innovative experiments using this technique to demonstrate that specific reaction products, of interest both in combustion chemistry and astrochemistry, can actually be formed from a specific single bimolecular reaction between a radical and a molecule, providing alternative, illuminating mechanisms not considered before by the communities. In particular, Prof. Kaiser has always emphasized the advantages of the CMB technique with respect to more traditional kinetics experiments, to sort out the nature of the primary products of an elementary bimolecular reaction.

But here, Prof. Kaiser resorts to a “chemical reactor” at 300 torr in which the residence time of the reactants is of **few hundreds of μ s** to identify the product of a bimolecular reaction! But the chemical reactor is a “**multi-collision environment**” ! Many secondary reactions can take place! The conclusion of the paper is that the products observed at $m/z=228$ correspond to two possible isomers: for 1/3 to 4-helicene (**p1**) and for 2/3 to 4-vinylpyrene (**p2**) on the basis of a two-contribution fit of the photoionization efficiency (PIE) curves for $m/z=228$, exploiting the PIE curves of the pure **p1** and **p2** species.

Here, I think that it would be interesting to see reported (at least in the SI) the PIE curves for also the two other possible products **p3** and **p4**. Are the curves for these two isomer products expected to be very different? Or rather are they very similar to those of **p1** and **p2**?

On page 6 of the manuscript the authors say: *It is vital to highlight that the PIE curves of the C18H12 isomers are specifically linked to each specific isomer underlining that the co-existence of additional isomers in the molecular beam except [4]-helicene and 4-vinylpyrene would significantly alter the shape of the PIE dramatically; therefore, alternative C18H12 isomers can be eliminated to contribute to $m/z = 228$. Consequently, we conclude that only [4]-helicene and 4-vinylpyrene account for the observed signal at $m/z = 228$ within our error limits.*

Have the p3 and p4 isomer products been excluded on the basis of their very different PIE curves? Eventually, these should be shown.

Prof. Kaiser has the ability to perform an experimental study of the bimolecular reaction in question under truly single-collision conditions in CMBs, because recently he has already demonstrated this capability by studying numerous reactions of PAH radicals with unsaturated hydrocarbons. These experiments would prove unambiguously whether for instance the products at mass 228 observed in the present study are actually the products **p1/p2** (corresponding to channels exoergic by about 240 kJ/mol) or rather the products **p3/p4** (channels exoergic by 15-28 kJ/mol) (see Fig. 4 in SI). I recognize that to discriminate between **p1** and **p2** would have been difficult in CMB experiments with hard electron ionization detection, and measurements of product efficiency curves, as can be done with tunable VUV synchrotron radiation, would be helpful.

Two considerations are in order here:

1) In the present work the potential energy surface for the relevant reaction has been calculated. Prof. Kaiser usually accompanies his combined experimental/theoretical studies with statistical calculations of the product branching ratios using the calculated potential surface. **This has not been done here.** But I think that **it would be interesting to see the statistical predictions of the branching ratios for not only the products p1 and p2, but also for the products p3, and p4.**

2) I notice that the very complex pathways leading to product **p1** (4-helicene) involve very high potential barriers for isomerization between the various intermediates, located at only -21, -26, and -41 kJ/mol from the reactant asymptote (141 kJ/mol from intermediate **[3]**, 36 kJ/mol from intermediate **[5]** and 132 kJ/mol from intermediate **[6]**) (see Figure 4). The other pathway leading to **p1** starting from intermediate **[4]** exhibits even higher barriers (up to -10 from the reactant asymptote, which corresponds to 167 kJ/mol from intermediate **[9]**). These barriers should be compared with the exoergicity (**-15 and -28 kJ/mol**) of the two direct pathways leading from intermediates **[3]** and **[4]** directly to products **p3** and **p4**, respectively (see Figure 4 in the SI).

a) **The authors should show, via statistical calculations, that the branching ratios of products p3 and p4 are actually negligible with respect to those of products p1 and p2, as it has been assumed here.** I have doubts about it.

b) The authors should also show that the PIE curves of the p3 and p4 products are not similar to those of products p1 and p2. I also doubt about this.

c) I would encourage Prof. Kaiser to tackle this reaction in CMB experiments and accompany the experimental work with statistical calculations!

Without this additional information and related comments the authors cannot conclude unambiguously, from the type of (limited) experiment and (limited) calculations that they done, that the elementary bimolecular reaction of 4-phenanthrenyl radical ($[C_{14}H_9]^\bullet$) with vinylacetylene (C_4H_4) at the temperature of 1450 K leads to 4-helicene + H (for 1/3) and to vinylpyrene + H (for 2/3).

In conclusion, I think that the idea behind this work is outstanding. The conclusions appear fascinating. But I find that the present experimental and theoretical work does not support unambiguously the conclusions. The paper may ultimately deserve publication in Nature Communications, after all above comments have been addressed in detail.

Response to Reviewers

Reviewer #1

In their manuscript, the authors address the "Gas Phase Synthesis of [4]-Helicene via a Versatile Vinylacetylene-Mediated Free Radical Ring Annulation". They present their finding as "A universal synthetic route to racemic helicenes". This is not at all what is demonstrated in the manuscript since only the obtention of benzo[c]phenanthrene, i.e. [4]helicene, is shown and there is no evidence at all that the same mechanism applies to longer helicenes. Indeed, regioselective aspects together with steric hindrance issues may be encountered.

The authors have clearly a strong expertise in the domains of PAHs and on the techniques they describe in this manuscript both experimentally (gas phase reactions, photoionization mass spectrometry) and theoretically (potential energy calculations). However, I found the manuscript very technical, both from the experimental and from the theoretical point of view, and not general enough to deserve publication in Nature Communications. I would rather suggest a more specific journal. Of course the same work on the case study carbo[6]helicene would have had a bigger impact.

Reply: We plan to carry out to perform similar experiments to form [5]- and [6]-helicenes in the future but this is beyond the scope of the present work. In our manuscript, we clearly stated that we present a proof of concept study on [4]-helicene, which in principle can form higher helicenes as well.

The authors mention "stereo selective reaction dynamics" when they discuss the "computational prediction of the formation of two structural isomers of C₁₈H₁₂ –" I think that they actually meant regioselective because the two isomers come from the cyclization at different positions.

Reply: Thank you. 'Stereo' has been changed to 'regio'.

The references given for the helicenes are not really appropriate. For example, the authors give references regarding catalysis or azahelicenes but more importantly there are many articles on helical PAHs, which, in my opinion, are more related to the present work.

Reply: We changed the paragraph.

Reviewer #2

The manuscript is very well written; in particular, the introductory page is excellent. The conclusions are fascinating. It sounds a wonderful story that may be true. But I have some concerns and few comments are therefore in order.

1) Only one typo spotted: in the Experimental Methods, 5 lines before the bottom of the section, "of 4- vinylpyrene" should be "or 4-vinylpyrene".

Reply: The typo has been corrected.

2) In the last five lines before the Conclusion the isomer products p3 and p4 are mentioned, but they are not depicted in Figure 4 of the main text. They are instead depicted in Figure 4 of the Supplementary Information, but this is not specified (so one cannot see where they are while

reading the main text). I suggest to include the pathways to isomers p3 and p4 also in Figure 4 of the main text.

Reply: They are now included in Figure 4.

3) My main concerns will actually start from the above mentioned five lines:

It should be noted that the initial covalently bound intermediates [3] and [4] can undergo hydrogen atom losses to form 1- and 4-phenanthrenyl-vinylacetylene isomers p3 and p4, respectively. These pathways are noticeable less favorable energetically than the formation pathways of [4]-helicene and 4-vinylpyrene – more so that in the prototype phenyl plus vinylacetylene reaction – but may play a role at high temperatures due to a favorable entropic factor.

Having said that I am very impressed by this manuscript, I have to say that I am also puzzled by several aspects of this study, and in particular by the approach used to reach the conclusions. We all know that the unambiguous proof that the bimolecular reaction $C_{14}H_9 + C_4H_4$ leads to $H + C_{18}H_{12}$ (4-helicene) can only come from experiments under single-collision conditions, as feasible using the crossed molecular beam (CMB) scattering technique. Prof. Kaiser has stressed this since the beginning of his career. Prof. Kaiser is one of the leading players in CMB experiments and has built his reputation by performing very nice and innovative experiments using this technique to demonstrate that specific reaction products, of interest both in combustion chemistry and astrochemistry, can actually be formed from a specific single bimolecular reaction between a radical and a molecule, providing alternative, illuminating mechanisms not considered before by the communities. In particular, Prof. Kaiser has always emphasized the advantages of the CMB technique with respect to more traditional kinetics experiments, to sort out the nature of the primary products of an elementary bimolecular reaction.

But here, Prof. Kaiser resorts to a “chemical reactor” at 300 torr in which the residence time of the reactants is of few hundreds of μs to identify the product of a bimolecular reaction! But the chemical reactor is a “multi-collision environment” ! Many secondary reactions can take place! The conclusion of the paper is that the products observed at $m/z=228$ correspond to two possible isomers: for 1/3 to 4-helicene (p1) and for 2/3 to 4-vinylpyrene (p2) on the basis of a two-contribution fit of the photoionization efficiency (PIE) curves for $m/z=228$, exploiting the PIE curves of the pure p1 and p2 species.

Here, I think that it would be interesting to see reported (at least in the SI) the PIE curves for also the two other possible products p3 and p4. Are the curves for these two isomer products expected to be very different? Or rather are they very similar to those of p1 and p2?

On page 6 of the manuscript the authors say: It is vital to highlight that the PIE curves of the $C_{18}H_{12}$ isomers are specifically linked to each specific isomer underlining that the co-existence of additional isomers in the molecular beam except [4]-helicene and 4-vinylpyrene would significantly alter the shape of the PIE dramatically; therefore, alternative $C_{18}H_{12}$ isomers can be eliminated to contribute to $m/z = 228$. Consequently, we conclude that only [4]-helicene and 4-vinylpyrene account for the observed signal at $m/z = 228$ within our error limits.

Have the p3 and p4 isomer products been excluded on the basis of their very different PIE curves? Eventually, these should be shown.

Prof. Kaiser has the ability to perform an experimental study of the bimolecular reaction in question under truly single-collision conditions in CMBs, because recently he has already demonstrated this capability by studying numerous reactions of PAH radicals with unsaturated hydrocarbons. These experiments would prove unambiguously whether for instance the products at mass 228 observed in the present study are actually the products p1/p2 (corresponding to channels exoergic by about 240 kJ/mol) or rather the products p3/p4 (channels exoergic by 15-28 kJ/mol) (see Fig, 4 in SI). I recognize that to discriminate between p1 and p2 would have been difficult in CMB experiments with hard electron ionization detection, and measurements of product efficiency curves, as can be done with tunable VUV synchrotron radiation, would be helpful.

Two considerations are in order here:

1) In the present work the potential energy surface for the relevant reaction has been calculated. Prof. Kaiser usually accompanies his combined experimental/theoretical studies with statistical calculations of the product branching ratios using the calculated potential surface. This has not been done here. But I think that it would be interesting to see the statistical predictions of the branching ratios for not only the products p1 and p2, but also for the products p3, and p4.

2) I notice that the very complex pathways leading to product p1 (4-helicene) involve very high potential barriers for isomerization between the various intermediates, located at only -21, -26, and -41 kJ/mol from the reactant asymptote (141 kJ/mol from intermediate [3], 36 kJ/mol from intermediate [5] and 132 kJ/mol from intermediate [6]) (see Figure 4). The other pathway leading to p1 starting from intermediate [4] exhibits even higher barriers (up to -10 from the reactant asymptote, which corresponds to 167 kJ/mol from intermediate [9]). These barriers should be compared with the exoergicity (-15 and -28 kJ/mol) of the two direct pathways leading from intermediates [3] and [4] directly to products p3 and p4, respectively (see Figure 4 in the SI).

a) The authors should show, via statistical calculations, that the branching ratios of products p3 and p4 are actually negligible with respect to those of products p1 and p2, as it has been assumed here. I have doubts about it.

b) The authors should also show that the PIE curves of the p3 and p4 products are not similar to those of products p1 and p2. I also doubt about this.

*c) I would encourage Prof. Kaiser to tackle this reaction in CMB experiments and accompany the experimental work with statistical calculations! Without this additional information and related comments the authors cannot conclude unambiguously, from the type of (limited) experiment and (limited) calculations that they done, that the elementary bimolecular reaction of 4-phenanthrenyl radical (*C14H9+•) with vinylacetylene (C4H4) at the temperature of 1450 K leads to 4-helicene + H (for 1/3) and to vinylpyrene + H (for 2/3).*

In conclusion, I think that the idea behind this work is outstanding. The conclusions appear fascinating. But I find that the present experimental and theoretical work does not support unambiguously the conclusions. The paper may ultimately deserve publication in Nature Communications, after all above comments have been addressed in detail.

Reply: First, we have synthesized the products **p3** and **p4** and measured their PIE curves. However, inclusions of any contributions from PIE curves from these isomers do not improve

the fit of the experimental PIE curve, which indicates that **p3** and **p4** can be produced only in minor amounts.

Second, to account for all comments of the referee, we have also carried out statistical RRKM-ME calculations of product branching ratios along with computational fluid dynamics simulations of the gas flow and kinetics in the microreactor. The results are described in detail in Supplementary Information.

Finally, we would like to note that, unfortunately, CMB experiments on this reaction are not feasible at this time since no molecular beam of the radical precursor can be prepared as of today.

Reviewer #3

This is an interesting paper and certainly deserves to be published in Nature Communication.

What are the major claims of the paper? Are they novel and will they be of interest to others in the community and the wider field? The authors claim that they identified a gas-phase reaction route towards 4-helicene. This is important and new for several reasons, including that until now only liquid-phase reaction mechanisms for 4-helicene synthesis exist.

Is the work convincing, and if not, what further evidence would be required to strengthen the conclusions?

Yes, given the current knowledge of gas-phase chemistry and with the theoretical evidence provided in this paper, the results are convincing. However, the discussion about the exclusion of different C₁₈H₁₂ isomers is rather weak. Again, I don't doubt their results, it's just that their arguments are pretty weak for excluding other conceivable C₁₈H₁₂ isomers.

Reply: As stated in the response to Reviewer #2, we included additional C₁₈H₁₂ isomers into consideration as potential reaction products both experimentally (via synthesis and measurements of their PIE curves) and theoretically (via statistical RRKM-ME calculations and computational fluid dynamics simulations of the gas flow and kinetics in the microreactor).

Do you feel that the paper will influence thinking in the field? Yes, I feel that people will include this reaction in their reaction mechanisms to explore the importance of this reaction.

Comment on the ability of a researcher to reproduce the work, given the level of detail provided: Yes, all necessary details are provided.

Before final publication, I would suggest the authors taking the additional suggestions into account:

Have the authors considered other potential pathways leading to the targeted intermediate. The mass spec at the low photon energies is rather misleading because many important intermediates (such as acetylene) are not present. What else is happening in the reactor? Where does the acetylene come from and what role does it play in the reaction network?

Reply: In our computational fluid dynamics simulations of the gas flow and chemical kinetics in the microreactor included in the revised version of the manuscript (Supplementary Information) we considered an extended chemical mechanism involving most important intermediates present in the reactor and secondary reactions. In particular, the kinetic mechanism included thermal

unimolecular decomposition of $C_{14}H_9Br$ and C_4H_4 , various product channels of the 4-phenanthrenyl ($C_{14}H_9$) + vinylacetylene (C_4H_4) reaction and H-assisted isomerizations among its primary $C_{18}H_{12}$ products, as well as H-assisted decomposition of C_4H_4 and reactions of $C_{14}H_9$ with acetylene (C_2H_2) and H atoms. Concerning acetylene, our previous work on the $C_6H_5 + C_4H_4$ reaction (Ref. 58) has shown that it is mostly formed in the $H + C_4H_4 \rightarrow C_2H_2 + C_2H_3$ reaction and the subsequent H loss from C_2H_3 . The proposed mechanism allowed us to account for practically all species experimentally observed in the mass spectrum.

Reviewers' comments:

Reviewer #1 (Remarks to the Author):

I am satisfied with the corrections made by the authors, although the introduction is still focused on chiral and configurationally stable helicenes and not really on benzophenanthrene derivatives, to which the molecule described in the manuscript corresponds.

All other corrections suggested were taken into account.

As a novel way of preparing PAH systems, I consider this manuscript suitable for publication in Nature Communications.

Reviewer #2 (Remarks to the Author):

Referee report for Nature Communications revised manuscript NCOMMS-18-21302

I recognize that the authors have made a considerable effort in terms of calculations/simulations to address some of the points I had brought up, as the additional details in the revised SI clearly show. However, I find the rebuttal of the authors to my comments/concerns to be rather weak and unsatisfactory. The reasons are outlined in the following.

1) Please note that in the SI, Table S4 is erroneously indicated as S3 (clearly a typo).

2) From the branching ratios reported in the SI Table S4 one sees that for the molar fractions of C₁₄H₉Br and C₄H₄ of 0.001 and 0.05, respectively, which are the values mimicking the experiment (see Methods-Experimental in the main text), the statistical calculations predict the following ratio of possible products:

p1	1
p2	29
p3	2.38
p4	2.36

This clearly indicates that formation of [4]-Helicene (Benzo(c)phenanthrene) (product p1) is minor under the experimental conditions, assuming that the model on which the statistical calculations are based is realistic. Clearly a concentration at the level of about 3% of p1 with respect to the other three possible isomer products is outside the experimental sensitivity.

3) I note that in the revision the SI Fig. 2 contains two more PIE curves of C18H12 isomers with respect to the SI of the original submission. Now, there are two very important points to comment about these results in regard to the implications they have in the data fitting/simulations.

(i) There are not units on the Y-axis. This means, as acknowledged in the SI text, that “absolute photoionization cross sections are not available for any of the considered C18H12 isomers”. Unfortunately this is a huge complication if one want to derive branching ratios by adding two or more of these curves.

(ii) The authors made a significant effort to synthesize all these possible C18H12 isomers and then to measure their PIE curves. However, a very important detail has not been specified. I infer that the PIE curves of SI Fig. 2 have been measured at room temperature, while they would be needed at 1450 K of the experiment. One would like to see the curves depicted in SI Fig. 2 measured at 1450 K and compared to those reported presently in the figure. I expect a “significant” red-shift of the curves. If the threshold of the PIE curves would exhibit a red shift (that can easily be of the order of several tenths of eV), the present approximate analysis would collapse.

There are two additional points of critics to the simulations reported in Fig. 3 of the main text: (i) as already alluded to above, one is the lack of knowledge of absolute photoionization cross sections of the contributing product isomers; (ii) the other is that, according to the statistical results of SI Table S4 (and commented above), the p3 and p4 isomer products should probably also be included.

(4) The authors have bypassed my main remark “(c) I would encourage Prof. Kaiser to tackle this reaction in CMB experiments and accompany the experimental work with statistical calculations!” saying that they don’t have a beam of 4-phenathrenyl radicals. Perhaps, with their ability to produce beams of this kind of radicals, they should have made the effort to achieve its production when addressing this interesting and challenging study. Likely, this would have prevented most of my comments.

(5) A final remark: at 300 torr of the experiment in the reactor, I would expect a significant collisional “stabilization” of some of the reaction intermediates (such as 4, 12, 13, 7, 11). The authors have not reported evidence of stabilized adducts/intermediates, because the signal they observe at $m/z=229$ is about that expected from the ^{13}C natural isotopic abundance. However, if stabilization would occur, I would not expect a significant amount of parent ion at $m/z=229$, not even by near threshold photoionization (actually we are about 2 eV above threshold, which may be a lot), considering their very high internal (ro-vibrational) energy content. This has not been commented on. If stabilization would occur, you would need CMB experiments to distinguish whether the signal at $m/z=228$ corresponds to a fragment of the adduct or to the parent of the heavy co-product of the H loss channel.

In conclusion, I am afraid that I have to convene with the doubts of Reviewer 1 about the suitability of this manuscript for publication in NATURE Comm., with the very significant addition (with respect

to Reviewer 1) that without the additional information (and related consideration) indicated in my comments above, the authors cannot conclude unambiguously and convincingly, from the type of limited experiments and approximate analysis that they have done, that the elementary bimolecular reaction of 4-phenanthrenyl radical ($[C_{14}H_9]^\bullet$) with vinylacetylene (C_4H_4) at the temperature of 1450 K leads for 33 \pm 10% to [4]-helicene + H and for 67 \pm 10% to vinylpyrene + H. In particular, if the reported statistical calculations and kinetic modeling are reliable, at the most one could expect a few percent of [4]-Helicene product as primary product under the experimental conditions, an amount impossible to quantify in this kind of experiment.

Reviewer #3 (Remarks to the Author):

The reviewers had commented on the need for additional work that would strengthen the authors' conclusions and arguments. In the meantime, the authors have performed additional experimental and theoretical work. Most of these new findings, which are now included in the manuscript (or the Supplementary Material), strengthen their paper and provide more evidence supporting the analysis of their data. In addition they also have addressed all of the other reviewers' comments satisfactorily. In my opinion, the findings and conclusions are supported by their presented data and that the paper is now acceptable for publication.

Reviewer #1

As a novel way of preparing PAH systems, I consider this manuscript suitable for publication in Nature Communications.

Thank you

Reviewer #2

However, I find the rebuttal of the authors to my comments/concerns to be rather weak and unsatisfactory.

We respectfully disagree with this comment. We addressed all comments extensively, synthesized new calibration compounds, and added significant new calculations/simulations. Most important, the referee report contains critical **factual errors**.

Please note that in the SI, Table S4 is erroneously indicated as S3 (clearly a typo).

We apologize for this mistake. This has been corrected.

From the branching ratios reported in the SI Table S4 one sees that for the molar fractions of C₁₄H₉Br and C₄H₄ of 0.001 and 0.05, respectively, which are the values mimicking the experiment (see Methods-Experimental in the main text), the statistical calculations predict the following ratio of possible products:

p1	1
p2	29
p3	2.38
p4	2.36

This clearly indicates that formation of [4]-Helicene (Benzo(c)phenanthrene) (product p1) is minor under the experimental conditions, assuming that the model on which the statistical calculations are based is realistic.

The referee is speculating, but does not provide any facts that our model is unrealistic.

Clearly a concentration at the level of about 3% of p1 with respect to the other three possible isomer products is outside the experimental sensitivity.

In the revised version of SI we clearly stated that “Branching ratios consistent with experiment can be obtained with initial 1% or 5% of C₁₄H₉Br and 5% of C₄H₄ in the molecular beam” and those branching ratios are presented in Table S3, whereas the referee picked in his argument the branching ratios computed for 0.1% of C₁₄H₉Br; the 0.1% was a leftover from the previous version of the manuscript. Moreover, we continue our discussion in SI stating that “... a direct comparison between the calculations and experiment is complicated by the fact that the absolute ionization cross sections are not available for any of the considered C₁₈H₁₂ isomers. If the absolute ionization cross section of **p1** significantly exceeds those of **p3** and **p4**, the contribution

of the latter two isomers to the experimental PIE curve is masked, whereas that of [4]-helicene is enhanced.” In the meantime, the fit of the experimental PIE curve of the ion counts **unambiguously** indicates the formation of [4]-helicene (**which represents the main and undeniable result of our work**) and a small to negligible contribution of **p3** and **p4**. Our consideration of the modeling results in SI shows how they can be reconciled with experiment.

Also, this comment of the referee is partially another **factual error**. The ion counts of [4]-helicene are ($33 \pm 10\%$) and of 4-vinylpyrene ($67 \pm 10\%$). These are outside the error limits. If we eliminate [4]-from the fit, the experimental PIE curve lacks intensity in the low energy section from the onset on; if we add isomers **p3** and **p4** to the fit, the intensity in the high photon energy range is elevated bringing the total ion counts outside the error limits.

We can alleviate the referee’s concern by making the following corrections in the main text:

Page 10: 4-phenanthrenyl radicals $[C_{14}H_9]^*$ were generated *in situ* via pyrolysis of the 4-bromophenanthrene precursor ($C_{14}H_9Br$) seeded in vinylacetylene/helium (5% C_4H_4 ; 95% He; Airgas) carrier gas at a pressure of 300 Torr in the entrance of the reactor. Accurate estimation of the concentration of $C_{14}H_9Br$ in the incipient molecular beam is difficult because the vapor pressure of this precursor is not known but the results of the computational fluid dynamics simulations and kinetic modeling (Supplementary Information) are consistent with experiment at the level of $C_{14}H_9Br$ concentration of 1-5%.

Page 6: After “The overall fit (black) consists of [4]-helicene ($33 \pm 10\%$) and of 4-vinylpyrene ($67 \pm 10\%$). It should be noted that inclusions of any contributions from 4-((*E*)-but-1-en-3-yn-1-yl)phenanthrene and 4-(but-3-en-1-yn-1-yl) phenanthrene do not improve the fit of the experimental data. This indicates that these isomers can be produced only at levels within the experimental error limits of the ion counts.”

We also included the following statement: “We emphasize here that the aforementioned contributions to the fit of the experimental PIE curve do not actually represent the product branching ratios since absolute ionization cross sections are not available for any of the considered $C_{18}H_{12}$ isomers. A more detailed discussion on the product branching ratios based on computational fluid dynamics (CFD) simulations coupled with kinetic modeling of chemical reactions inside the reactor is given in the SI.”

3) I note that in the revision the SI Fig. 2 contains two more PIE curves of $C_{18}H_{12}$ isomers with respect to the SI of the original submission. Now, there are two very important points to comment about these results in regard to the implications they have in the data fitting/simulations.

(i) There are not units on the Y-axis. This means, as acknowledged in the SI text, that “absolute photoionization cross sections are not available for any of the considered C₁₈H₁₂ isomers”.

We changed the units to ‘ion counts’.

Unfortunately this is a huge complication if one want to derive branching ratios by adding two or more of these curves.

The major result of the present work is the identification of [4]-helicene and the elucidated pathways. This can be achieved without providing branching ratios. In the response to the previous comment, we clearly distinguished between the contributions to the fit and branching ratios and included this clarification in the revised version in p. 6. To provide absolute branching ratios, absolute photoionization cross sections are required; no group on this planet is able to accurately calculate or measure these cross sections experimentally thus far.

(ii) The authors made a significant effort to synthesize all these possible C₁₈H₁₂ isomers and then to measure their PIE curves. However, a very important detail has not been specified. I infer that the PIE curves of SI Fig. 2 have been measured at room temperature, while they would be needed at 1450 K of the experiment. One would like to see the curves depicted in SI Fig. 2 measured at 1450 K and compared to those reported presently in the figure. I expect a “significant” red-shift of the curves. If the threshold of the PIE curves would exhibit a red shift (that can easily be of the order of several tenths of eV), the present approximate analysis would collapse.

Unfortunately, this is a **factual error**. We clearly stated in the manuscript that these calibration curves are recorded under identical physical conditions (pressure, temperature) as the real experiment. Also, we have a supersonic expansion. This – as evidenced in the manuscript – cools the molecules. **Therefore, we state that these curves have been recorded at 1400 K.**

To make this clear, we have added the following statement in the SI related to the calibration PIEs:

These PIE calibration curves of helium-seeded C₁₈H₁₂ isomers were newly recorded in this work under identical conditions (pressure, temperature) as the real experiments and are shown as black along with the error limits (grey area).

There are two additional points of critics to the simulations reported in Fig. 3 of the main text: (i) as already alluded to above, one is the lack of knowledge of absolute photoionization cross sections of the contributing product isomers.

See our reply above. It is fair to raise this issue again and again as a challenge for future experimental and/or theoretical work, but not nice to hinder publication of this work, which does

not claim that product branching ratios were actually measured experimentally. The mechanistical study was the main goal of this work.

(ii) the other is that, according to the statistical results of SI Table S4 (and commented above), the p3 and p4 isomer products should probably also be included.

This is once again a **factual error**. The referee again is using ‘probably’. We stated clearly in the revised version that these isomers have no contribution to the PIE curve. An inclusion of these curves **cannot fit the experimental data**. This is an undeniable fact.

(4) The authors have bypassed my main remark “c) I would encourage Prof. Kaiser to tackle this reaction in CMB experiments and accompany the experimental work with statistical calculations!” saying that they don’t have a beam of 4-phenathrenyl radicals. Perhaps, with their ability to produce beams of this kind of radicals, they should have made the effort to achieve its production when addressing this interesting and challenging study. Likely, this would have prevented most of my comments.

We did not bypass this comment. We stated that it is a fact that it is currently not feasible to produce a supersonic beam of 4-phenathrenyl radicals **at concentrations high enough to achieve scattering signal**. Unfortunately, making a beam does not always produce a discernable signal. The referee cannot request something which is not feasible (5) A final remark: at 300 torr of the experiment in the reactor, I would expect a significant collisional “stabilization” of some of the reaction intermediates (such as 4, 12, 13, 7, 11). The authors have not reported evidence of stabilized adducts/intermediates, because the signal they observe at $m/z=229$ is about that expected from the ^{13}C natural isotopic abundance. However, if stabilization would occur, I would not expect a significant amount of parent ion at $m/z=229$, not even by near threshold photoionization (actually we are about 2 eV above threshold, which may be a lot), considering their very high internal (ro-vibrational) energy content. This has not been commented on. If stabilization would occur, you would need CMB experiments to distinguish whether the signal at $m/z=228$ corresponds to a fragment of the adduct or to the parent of the heavy co-product of the H loss channel.

This is yet another factual error. The **inlet** pressure at the reactor is 300 Torr; as evidenced in the manuscript (Supplementary Figure 5), the pressure **inside** the reactor ranges from 40 Torr to 5 Torr. Under these conditions there is **no stabilization**.

In conclusion, I am afraid that I have to convene with the doubts of Reviewer 1 about the suitability of this manuscript for publication in NATURE Comm., with the very significant addition (with respect to Reviewer 1) that without the additional information (and related consideration)

In fact, Referee 1 recommends publication in Nat. Com.

indicated in my comments above, the authors cannot conclude unambiguously and convincingly, from the type of limited experiments and approximate analysis that they have done,

We in fact demonstrated in the manuscript that our conclusions, e.g. the formation of [4]-helicene in the considered reaction under our experimental conditions are solid and undeniable. Likewise, the experiments are not limited, but, unfortunately this referee demonstrated a limited understanding of the experiments we conducted.

In particular, if the reported statistical calculations and kinetic modeling are reliable,

The referee has not provided a single **fact** that the calculations are unreliable.

at the most one could expect a few percent of [4]-Helicene product as primary product under the experimental conditions,

This is correct – as stated in the manuscript. However, the referee does not understand – as stated in the manuscript – that this process involves primary and secondary reactions (which were modeled!). Secondary reactions can convert – as demonstrated in the manuscript, higher energy isomers to [4]-helicene.

an amount impossible to quantify in this kind of experiment.

Quantifying product branching ratios was not and is not stated as the main goal or the main result of this experiment; the formation of [4]-helicene and the mechanism leading to it are! By the way, quantification of product branching ratios in CMB experiments proposed by the referee is even more challenging and cannot be conducted for this system based on the experience of one of the corresponding authors (RIK).

Also we would like to highlight that the conclusions compile that [4]-helicene can be formed – as elucidated in our experimental and theoretical study – via two pathways. This presents an objective and correct main conclusion of our work.

Also, this comment of the referee is another **factual error**. The **ion counts** of [4]-helicene are ($33 \pm 10\%$) and of 4-vinylpyrene ($67 \pm 10\%$). These are **outside the error limits**.

The main goal was and still is the mechanistical work how [4]-helicene can be formed. This has been accomplished:

Consequently, PAHs as complex as [4]-helicene ($C_{18}H_{12}$) can be formed via a *bimolecular reaction* in circumstellar envelopes of dying carbon stars. However, in combustion flames such as of benzene, where free hydrogen atoms are ubiquitous, secondary reactions via a hydrogen atom assisted isomerization of the (*E*)-4-(but-1-en-3-yn-1-yl)phenanthrene isomer likely also play an important role in the formation of [4]-helicene.

Reviewer #3

In my opinion, the findings and conclusions are supported by their presented data and that the paper is now acceptable for publication.

Thank you.